# Dual Hedgehog/GLI1 and PI3K/Akt/mTOR Targeting Possesses Higher Efficacy to Inhibit T-Cell Acute Lymphoblastic Leukemia Growth

**DOI:** 10.3390/cells14241972

**Published:** 2025-12-11

**Authors:** Marica De Chiara, Mariaconcetta Sicurella, Mattia Melloni, Ilaria Conti, Luca Maria Neri

**Affiliations:** 1Department of Translational Medicine, University of Ferrara, 44121 Ferrara, Italy; marica.dechiara@unife.it (M.D.C.); ilaria.conti@unife.it (I.C.); 2Department of Environmental Sciences and Prevention, University of Ferrara, 44121 Ferrara, Italy; scrmcn@unife.it; 3Department of Chemical, Pharmaceutical, and Agricultural Sciences, University of Ferrara, 44121 Ferrara, Italy; 4LTTA-Electron Microscopy Center, University of Ferrara, 44121 Ferrara, Italy

**Keywords:** T-cell acute lymphoblastic leukemia (T-ALL), Hedgehog and PI3K/Akt/mTOR signaling, synergistic drug combination

## Abstract

While the PI3K/Akt/mTOR pathway is a well-established drug target in T-cell acute lymphoblastic leukemia (T-ALL), the contribution of the Hedgehog (Hh) pathway in T-ALL malignancy remains poorly defined. We investigated the effects of pharmacological inhibition of key signaling nodes in these pathways using T-ALL cell lines (Jurkat, Molt-4, DND-41, and ALL-SIL). Cells were treated with the Gli1 inhibitor Gant-61, the Smoothened inhibitors GDC-0449 and Glasdegib, the Akt inhibitor MK-2206, and the mTOR inhibitor RAD001, both alone and in combination. Analyses of cell viability, cell cycle progression, apoptosis, autophagy, protein expression, and in situ intracellular distribution revealed potent cytotoxic activity of Gant-61 and MK-2206, while Smo and mTOR inhibitors showed limited efficacy. Combined Gli1 and Akt inhibition induced synergistic suppression of proliferation, enhanced G0/G1 arrest, increased apoptosis, and promoted autophagy, accompanied by reduced nuclear Gli1 and decreased Akt phosphorylation. These findings demonstrate a functional interaction between Hh/Gli1 and PI3K/Akt pathways in T-ALL and identify Gli1 as a critical, druggable node. Dual targeting of Gli1 and Akt represents a potential therapeutic strategy to overcome resistance and improve treatment outcomes in T-ALL.

## 1. Introduction

T-cell acute lymphoblastic leukemia (T-ALL) accounts for approximately 12–15% of all pediatric ALL cases and is an aggressive hematological disorder characterized by genetic abnormalities and microenvironmental factors leading to aberrant proliferation of T-cell precursors [1,2]. Alterations that contribute to the pathogenesis of T-ALL involve deregulation of different signal transduction pathways that control several biological processes, such as cell growth, survival, apoptosis, and differentiation. One of these is the PI3K/Akt/mTOR pathway that has been extensively examined in the field of T-ALL targeted therapy. In contrast, the role of the Hedgehog (Hh) pathway in T-ALL remains to be better explored. Hh is a signaling pathway involved in embryogenesis and tissue homeostasis, and it is impaired in different forms of cancer [3,4]. The canonical Hh signaling pathway is triggered by the interaction between one of the specific secreted ligands—Sonic Hedgehog (SHH), Desert Hedgehog (DHH), or Indian Hedgehog (IHH)—with the transmembrane receptor Patched (PTCH), which removes its inhibitory action on another transmembrane receptor, Smoothened (Smo). Once activated, Smo propagates the signal to the Gli family of transcription factors (Gli1, Gli2, and Gli3), which, free from the binding to their negative regulator Suppressor of Fused (SUFU), translocate from the cytoplasm into the nucleus. Gli1 and Gli2 function mainly as transcriptional activators, while Gli3 acts mainly as a transcriptional repressor [5]. Gli proteins are indeed transcriptional effectors of Hh signaling, and they interact with DNA by zinc finger domains (containing binding-DNA zinc fingers) [6]. All four T-ALL cell lines employed in this study—Jurkat, Molt-4, DND-41, and ALL-SIL—have been reported in the literature to have constitutive aberrant activation of the PI3K/Akt pathway, although through distinct mechanisms [7,8]. Previous studies have demonstrated that GLI1 and GLI2 are expressed and transcriptionally active in several T-ALL models, including Jurkat and MOLT-4 cells, indicating their involvement in leukemic cell survival and proliferation [9,10].

Crosstalk between Hedgehog and PI3K/Akt/mTOR signaling pathways, which can sustain proliferation and chemoresistance in T-cell leukemias, has been reported [11]. Emerging evidence supports a complex functional interaction between the Hh/Gli and PI3K/Akt/mTOR pathways in several cancer types, such as melanoma, prostate cancer, glioma, and hematologic malignancies [12,13,14]. This crosstalk modulates the expression, stability, nuclear localization, and transcriptional activity of Gli1, thereby promoting tumorigenesis and contributing to chemoresistance [12].

To assess the potential synergistic action of the two pathways, inhibitors targeting the Hedgehog or PI3K/Akt/mTOR pathway were employed. It has been previously demonstrated that GANT-61 has a high specificity for GLI1/GLI2, and its mechanism of action was also well characterized [15,16]. Similarly, Vismodegib (GDC-0449) and Glasdegib, were reported to be selective inhibitors of the Smo protein of the Hh pathway [17,18,19]. In many reports, MK-2206 has been described as a selective allosteric inhibitor of AKT1/2/3, preventing phosphorylation at both T308 and S473 and potently inhibiting downstream pro-survival signaling [14,17,20,21,22,23,24,25]. These inhibitors were selected based on their previously reported efficacy in hematological malignancies to explore their potential synergistic effects blocking the Hedgehog and PI3K/Akt/mTOR pathways simultaneously [26,27,28,29,30,31,32]. We observed that the combined inhibition of Hh and PI3K/Akt/mTOR signaling exerts a synergistic anti-leukemic effect, thereby underscoring the potential of this therapeutic strategy in T-ALL.

## 2. Materials and Methods

### 2.1. Materials

Fetal Bovine Serum (FBS; Euroclone) and RPMI 1640 cell culture medium were purchased from Euroclone (ECS0180). GDC-0049, Gant-61, Glasdegib, MK-2206, and RAD-001 (MedChem Express) were supplied pre-dissolved in dimethyl sulfoxide (DMSO) at a concentration of 10 mM and were subsequently diluted in RPMI 1640 to achieve a final DMSO concentration of 0.2%.

### 2.2. Cell Cultures

The human T-ALL cell lines Jurkat, Molt-4, DND-41, and ALL-SIL were obtained from DSMZ. Jurkat and DND-41 cells were cultured in RPMI 1640 supplemented with 10% FBS, 1% penicillin–streptomycin (10,000 U/mL; Euroclone-Milan, Italy, ECB3001D), and 1% L-glutamine (200 mM in 0.85% NaCl; Lonza-Euroclone Italy, BE17-605E). Molt-4 and ALL-SIL cells were cultured in RPMI 1640 supplemented with 20% FBS, 1% penicillin–streptomycin, and 1% L-glutamine. All four cell lines were maintained at 37 °C in a humidified atmosphere containing 5% CO_2_.

### 2.3. Cell Viability Assay

Cytotoxic effects of the pharmacological inhibitors were assessed using the Cell Counting Kit-8 (CCK-8; MedChem Express-DBA Milan, Italy, HY-K0301). Briefly, 2 × 10^5^ cells per well were seeded in 96-well plates and treated with increasing concentrations of the compounds for 24, 48, or 72 h. At the end of each treatment period, CCK-8 solution was added according to the manufacturer’s protocol, and plates were incubated for 4 h at 37 °C. Absorbance was then recorded using an Infinite M Plex plate reader (Tecan-GmbH Untersbergstr, Salzburg, Austria).

### 2.4. Cell Cycle, Apoptosis and Autophagy Assays

Cell cycle distribution was assessed using the Muse Cell Cycle Kit (CYTEK-MCH100106- Prodotti Gianni-Italy). Following 24 h exposure to IC_50_ concentrations of Gant-61 and MK-2206, administered alone or in combination, cells were fixed in 70% ethanol and incubated overnight at 4 °C. After washing with PBS, cells were resuspended in Muse Cell Cycle reagent and analyzed using the Muse Cell Analyzer (Merk-Millipore- Darmstadt, Germany).

Early and late apoptosis was evaluated with the Muse Annexin V & Dead Cell Kit (CYTEK-MCH100105-Prodotti Gianni-Italy). Cells treated with IC_50_ concentrations of Gant-61 and MK-2206, alone or in combination for 24 h, were collected, incubated with Muse Annexin V & Dead Cell reagent for 20 min at room temperature, and analyzed using the Muse Cell Analyzer.

Autophagic activity, indicated by LC3-II, was monitored using the Muse Autophagy LC3-antibody-based KIT (LUMCH200109 Prodotti Gianni-Italy,). Cells were prepared according to the manufacturer’s instructions and analyzed with the Muse Cell Analyzer. All experiments were performed in triplicate.

### 2.5. Western Blotting Analysis

Proteins from lysed leukemic cells were quantified using the BCA assay, separated on 7.5%, 10%, or 12% SDS-PAGE gels, and transferred onto nitrocellulose membranes. Membranes were blocked for 1 h at room temperature in TBS containing 0.1% Tween-20 and 5% nonfat milk, then incubated overnight at 4 °C with primary antibodies: Akt (1:1000, 9272S), phospho-Akt (Ser473, 1:1000, 4060S), GSK3β (1:1000, 9315S), phospho-GSK3β (Ser9, 1:1000, 9336S), p70S6K (1:1000, 9202), phospho-p70S6K (1:1000, 9205), and Gli1 (1:1000, 2534S). Following washes, membranes were incubated with HRP-conjugated secondary antibodies (anti-rabbit or anti-mouse, 1:3000; Cell Signaling-Euroclone Milan, Italy) and visualized using enhanced chemiluminescence (WBKLS0500 Merk-Millipore- Germany,) on the iBright 1500 system (Invitrogen, Thermo Fisher Scientific- Milan, Italy). β-actin (1:20,000; Thermo Scientific) was used as a loading control. Densitometric analysis was performed only for proteins that exhibited labeling-related modifications. Band intensities were normalized to β-actin and expressed as fold change relative to the control. Data are presented as mean ± SD from three independent experiments.

### 2.6. Immunofluorescence Assay

Following 6–24 h treatment with the inhibitors Gant-61 and MK-2206, leukemic cells were harvested and fixed with 4% paraformaldehyde (PFA) on poly-L-lysine-coated coverslips to ensure accurate adhesion. Non-specific binding sites were blocked with 1% BSA for 1 h at 37 °C. Cells were then incubated with primary antibodies: Gli1 (1:100, SAB5700740, Sigma-Aldrich-Darmstadt, Germany); phospho-Akt (Ser473, 1:400, 4060S Cell Signaling); and LC3A/B (1:100, 4108S Cell Signaling) for 1 h at 37 °C, followed by incubation with the corresponding secondary antibodies for an additional hour at 37 °C. Nuclei were counterstained with DAPI, and samples were dehydrated through a graded ethanol series before mounting. Coverslips were then examined using a Nikon Upright Fluorescence Microscope (Eclipse Ci-S/Ci-L-Nikon Europe, Amsterdam, The Netherlands).

### 2.7. Evaluation of Drug Combination

The combined effect of MK-2206 and Gant-61 was assessed using Compusyn software, which calculates a combination index (*CI*) based on the following equation:$$CI=\frac{D_{1}}{D_{1X}}+\frac{D_{2}}{D_{2X}}$$
where $D_{1}$ and $D_{2}$ are the concentrations of drugs used in combination that inhibit x% of cell viability, and $D_{1X}\;$ and $D_{2X}\;$ are the concentrations of the individual drugs that achieve the same effect. These concentrations were determined for each experiment and for each combination at a constant ratio of Gant-61 to MK-2206. According to this method, the drug combination is considered synergistic when *CI* < 1, additive when *CI* = 1, and antagonistic when *CI* > 1.

### 2.8. Statistical Analysis

Data are presented as mean ± SD and were analyzed for statistical significance using Student’s *t*-test and two-way ANOVA—multiple comparisons, as appropriate, with GraphPad Prism 8 software. Differences were considered statistically significant at *p* < 0.05.

## 3. Results

### 3.1. Cytotoxicity of Hedgehog and PI3K/Akt/mTOR Pathways Inhibitors in T-ALL Cell Lines

Gant-61 (Gli1 inhibitor), GDC-0049, and Glasdegib (Smo inhibitors) were administered one-by-one for 72 h (Figure 1A) to verify whether they exert cytotoxic activity on the following four T-ALL cell lines: Jurkat, Molt-4, DND-41, and ALL-SIL. The rates of cell survival were measured by CCK8 assay. Graphs showed that all four T-ALL cell lines were sensitive to Gant-61 to a different extent, but they were not sensitive to GDC-0049 and Glasdegib. In addition to MK-2206, we also tested the mTORC1 inhibitor RAD-001 (Everolimus), which also targets the PI3K/Akt/mTOR pathway, but since it acts on mTORC1, it targets an enzyme downstream of MK-2206, i.e., Akt.

MK-2206 (Akt inhibitor) and RAD-001 (Everolimus, mTORC1 inhibitor) were evaluated after 24 and 48 h of treatment (Figure 1B). MK-2206 showed cytotoxic effects already after 24 h of treatment with different degrees of efficacy. Of note, Jurkat and Molt-4 cell lines showed the highest sensitivity to the drugs employed. RAD-001 did not show the same efficacy as MK-2206 and did not reach a valuable IC_50_ (Table 1).

### 3.2. Combined Treatment with Gli1 Inhibitor and Akt Inhibitor in Jurkat and Molt-4 Gant-61 and MK-2206 Show Synergistic Effect on Jurkat and Molt-4 Cells

In the combination analysis, we selected Jurkat and Molt-4 for combined treatment since they showed the highest sensitivity to drug exposure. To avoid an excess of cell mortality, i.e., to have enough samples to analyze, we administered at the IC50 concentration previously obtained, Gant-61 for 72 h and MK-2206 only for the last 24 h at a constant ratio of 1:10 or 1:15 for Jurkat or Molt-4. Both in Jurkat and Molt-4 cells, at all the concentrations employed, a relevant synergy was always observed. For subsequent assays conducted on both cell lines under investigation, new IC_50_ values were used for the MK-2206 inhibitor, as it was added 48 h after seeding. This event was observable at the highest concentrations in Jurkat cells, whereas it was observable at the lowest concentrations in Molt-4 cells (Figure 2).

### 3.3. Gant-61 and MK-2206 Modify Gli1 Cellular Localization and p-Akt Phosphorylation in Jurkat and Molt-4 Cells

In untreated Jurkat and Molt-4 cells, fluorescent labeling of both Gli1 and p-Akt was observable mainly in the cytoplasm and to a lesser extent in the nuclear region. Fluorescence appears as both homogeneous staining and discrete dots distributed throughout the cytoplasm and around the nucleus (Figure 3A,B). Gli1 signal decreased in the cytoplasm as well as in the nuclear region in both cell lines when Gant-61 or MK-2206 were administered alone for 6 h (Figure 3A,B). Akt phosphorylation staining was slightly reduced in both cell compartments after Gant-61 exposure, whereas it was significantly decreased, especially at the nuclear level, when MK-2206 was employed. After 6 h of both drugs administration, Gli1 labeling was decreased either in the cytoplasm or in the nuclear area. At variance, p-Akt showed the fluorescent signal further decreased when compared with MK-2206 used alone and exclusively located in the cytoplasm. Interestingly, 24 h of combined treatment not only additionally reduced fluorescence in the cytoplasmic compartment but also fully abolished fluorescent staining in the nuclear region.

### 3.4. Synergistic Effect of Gant-61 and MK-2206 on Gli1 Expression and p-Akt and Its Substrates Phosphorylation

Similarly, as previously described, Gant-61 was administered to cells for 72 h at 19 or 20 μM, corresponding to the IC_50_ value of Jurkat or Molt-4 cells, respectively. In the last 24 h of treatment, MK-2206 was added at scaled concentrations up to that corresponding to the IC_50_ value for each cell line. We first examined Gli1 and p-Akt, together with its total protein expression. In Jurkat cells, Gli1 expression was reduced by Gant-61 and decreased progressively when MK-2206 increased. When the two drugs were administered together, Gli1 expression was further reduced according to drug concentration increase. p-Akt was progressively reduced linearly when two drugs were administered together. At the highest concentrations, no labeling was observable. Western blotting of two Akt substrates, p-GSK3β and p-p70S6K, showed the most relevant downregulation at the highest Gant-61 plus MK-2206 concentrations. In Molt-4 cells, Gli1 expression reduction was observable already at the concentration of 0.2 μM MK-2206. The downregulation after combined treatment was more pronounced when compared to Jurkat cells, thus strengthening the synergistic effect (Figure 4A,B). Whereas expression of p-Akt did not seem affected by Gant-61 administration, in Molt-4 cells, drug combination showed an increased inhibition of Akt phosphorylation when compared with single-drug cell exposure. A similar behavior was observable in the two Akt-phosphorylated substrates, GSK3β and p70S6K. Total forms were unchanged upon drug treatments.

### 3.5. Cytofluorimetric Analysis of Apoptosis and Cell Cycle Gant-61 and MK-2206 Increase G0/G1 Phase of Cell Cycle and Apoptosis in a Synergistic Way

Cell cycle distribution, apoptosis, and autophagy were analyzed by flow cytometry. Cell cycle distribution showed an increase in the G0/G1 phase and a decrease in the S and G2/M phases detectable already after single treatments but more evident after double drug exposure of cells (Figure 5A,B). Apoptosis was developed at the highest degree when Gant-61 and MK-2206 were administered together in Jurkat and Molt-4 cells (Figure 6A,B).

### 3.6. Autophagy Is Enhanced by the Combined Treatment with Gant-61 and MK-2206 in Jurkat and Molt-4 Cells

Autophagy was investigated as a further mechanism of cell death by means of immunofluorescence of LC3A/B in situ expression. The antibody recognized the non-cleaved form of the protein. The strong LC3A/B signal of the uncleaved protein indicated the absence of autophagy in untreated Jurkat (Figure 7A) and Molt-4 cells (Figure 7B). The labeling was almost homogeneously distributed throughout the entire cytoplasm. When Gant-61 or MK-2206 was administered for 6 h, LC3A/B expression decreased very significantly and became nearly completely absent in their combined treatment in both cell lines. In these samples, fluorescence staining was condensed in brilliant spots located at the periphery of the cytoplasm. Comparable staining was observable in 18 h Gant-61- and MK-2206-treated samples.

## 4. Discussion

This study demonstrates that the simultaneous inhibition of the Hedgehog (Hh) and PI3K/Akt/mTOR signaling pathways exerts a potent and synergistic cytotoxic effect in T-cell acute lymphoblastic leukemia (T-ALL) cells, highlighting a functional interdependence between these signaling cascades in maintaining leukemic cell survival. Among the Hedgehog pathway inhibitors tested, Gant-61, a direct inhibitor of the transcription factor Gli1, displayed pronounced cytotoxic activity across all T-ALL cell lines examined. In contrast, smoothened (Smo) antagonists, such as GDC-0049 and Glasdegib, were largely ineffective, suggesting that Gli1 activation in T-ALL primarily occurs through non-canonical, Smo-independent mechanisms. These observations align with previous evidence showing that Gli1 can be transcriptionally or post-translationally activated by oncogenic signaling through PI3K/Akt/mTOR or MAPK pathways, thereby bypassing canonical Hedgehog components [13,16,32,33,34,35]. Evidence from medulloblastoma and basal cell carcinoma indicates that resistance to Smo inhibitors often arises due to downstream mutations or alternative Gli1/2 activation. Therefore, strategies that directly inhibit Gli1 or modulate its regulators, such as histone deacetylases or non-canonical kinases, may represent a more effective approach for sustained tumor suppression [36,37]. Consistent with these observations, our immunofluorescence analysis revealed that combined treatment with Gant-61 or/and the allosteric Akt inhibitor MK-2206 markedly suppressed Gli1 nuclear localization and, consequently, its transcriptional activity. This was accompanied by a pronounced reduction in Akt phosphorylation, indicating reciprocal pathway inhibition [38,39]. It has been reported that Akt activity promotes Gli1 stabilization and nuclear translocation, while Gli1 in turn sustains Akt signaling with a crosstalk mechanism previously described in other malignancies [40,41].

Pharmacokinetic studies on MK-2206 have shown that plasma concentrations in the low micromolar range, comparable to those used in our experiments, are achievable in vivo and have been safely administered in clinical trials [20,42].

Similarly, although GANT-61 has been used primarily in preclinical settings, the effective concentrations employed in this study still fall within the pharmacologically achievable range in vivo [43,44]. Therefore, the drug concentrations we used in vitro can be a good reference for their translational applications.

The dual blockade of these pathways likely disrupts compensatory survival circuits that allow leukemic cells to persist under single-agent treatment. In support of this, co-treatment induced increased G0/G1 cell-cycle arrest, apoptosis, and autophagy when compared with single treatment [45,46]. The enhanced autophagic response suggests that dual inhibition amplifies cytotoxicity and highlights the therapeutic potential of concurrently targeting Gli1 and Akt in T-ALL [10] and represents a promising strategy to overcome intrinsic resistance mechanisms with robust anti-leukemic effects [47]. Previous observations reported that combined inhibition of the Hedgehog and PI3K/Akt/mTOR pathways resulted in enhanced cytotoxicity in some T-ALL cell lines. Hou et al. [48] demonstrated that simultaneous block of these signaling cascades synergistically reduced proliferation and increased apoptosis in leukemia and solid tumor cells. Our present study introduces new findings, explores different cell lines, new drugs, and analyzes new cell processes (autophagy and cell cycle) and new molecules of the signaling pathway with Western blotting and in situ immunofluorescence, giving strength to the concept that dual targeting of Gli1 and Akt represents a mechanistically and potentially translatable therapeutic approach in this aggressive leukemia subtype [49]. The genetic background of leukemia is also likely to impact drug response. Mutations in PI3K/Akt pathway genes or alterations in TP53 or SUFU may influence the degree of Gli1 activation and thus modulate sensitivity to pathway inhibition [11]. Stratifying T-ALL patients according to their mutational profile could therefore help to identify subsets most likely to benefit from combined targeting of Gli1 and Akt. Future studies employing primary cells or patient-derived xenograft (PDX) models may further confirm the therapeutic efficacy of GANT-61 and MK-2206 co-administration.

The concurrent targeting of Gli1 and Akt pathways represents a rational and promising strategy to overcome intrinsic or acquired resistance associated with Hedgehog inhibition, particularly in leukemias characterized by non-canonical Hedgehog activation [20,30,48,49].

## Figures and Tables

**Figure 1 cells-14-01972-f001:**
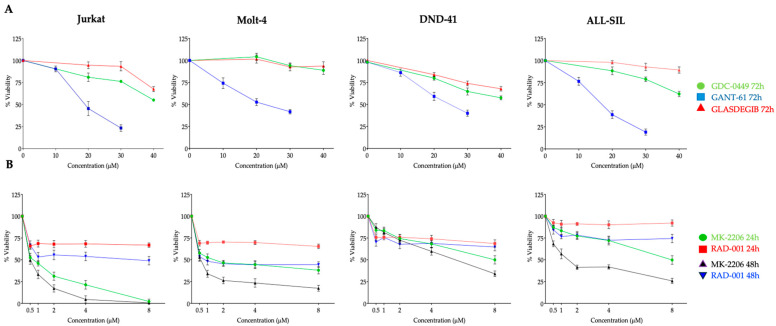
Effects of Hedgehog and Akt/mTOR pathway inhibitors on T-ALL cell viability. Dose–response curves showing the effects of Hedgehog pathway inhibitors GDC-0449, Gant-61, and Glasdegib on cell viability in Jurkat, Molt-4, DND-41, and ALL-SIL T-ALL cell lines after 72 h of treatment (**A**); in (**B**) dose–response curves showing the effects of Akt inhibitor MK-2206 and mTOR inhibitor RAD-0001 (Everolimus) on cell viability of the same T-ALL cell lines after 24 and 48 h of treatment. Cell viability was measured using a colorimetric assay and expressed as a percentage relative to untreated control cells. Data represent SD from at least three independent experiments. *p*-value < 0.05.

**Figure 2 cells-14-01972-f002:**
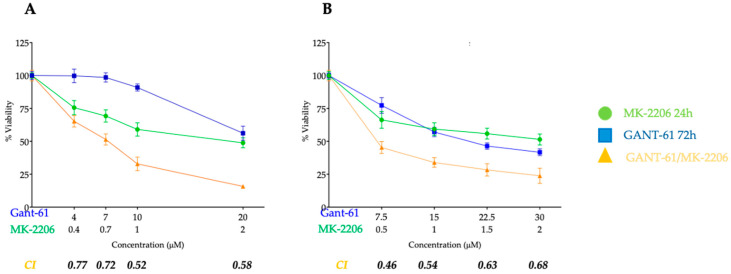
Combined treatment with Gant-61 and MK-2206 enhances cytotoxicity in T-ALL cell lines. Dose–response curves showing the effects of single or combined Gant-61 (72 h) and MK-2206 (24 h) treatment, and their combination, on the viability of Jurkat (**A**) and Molt-4 (**B**) T-ALL cell lines. Cell viability was measured using a colorimetric assay and expressed as the percentage of viable cells relative to untreated controls. The Combination Index (CI) values were calculated using Compusyn Software, with CI < 1 indicating synergistic effects. Data are presented as the mean ± SD of at least three independent experiments. *p*-value < 0.05.

**Figure 3 cells-14-01972-f003:**
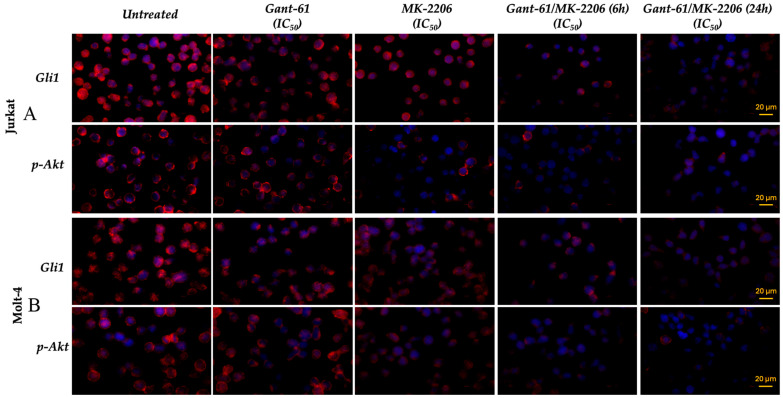
Immunofluorescence analysis of Gli1 and phosphorylated Akt (p-Akt) expression in Jurkat (**A**) and Molt-4 (**B**) cells. Cells were treated for 6 h with Gant-61 (IC_50_; 19 μM in Jurkat and 20 μM in Molt-4) or MK-2206 (IC_50_; 2 μM in Jurkat and 2.8 μM in Molt-4), alone or in combination for 6 and 24 h. Nuclei were stained with DAPI (blue). The scale bar is indicated in the last image of the panel.

**Figure 4 cells-14-01972-f004:**
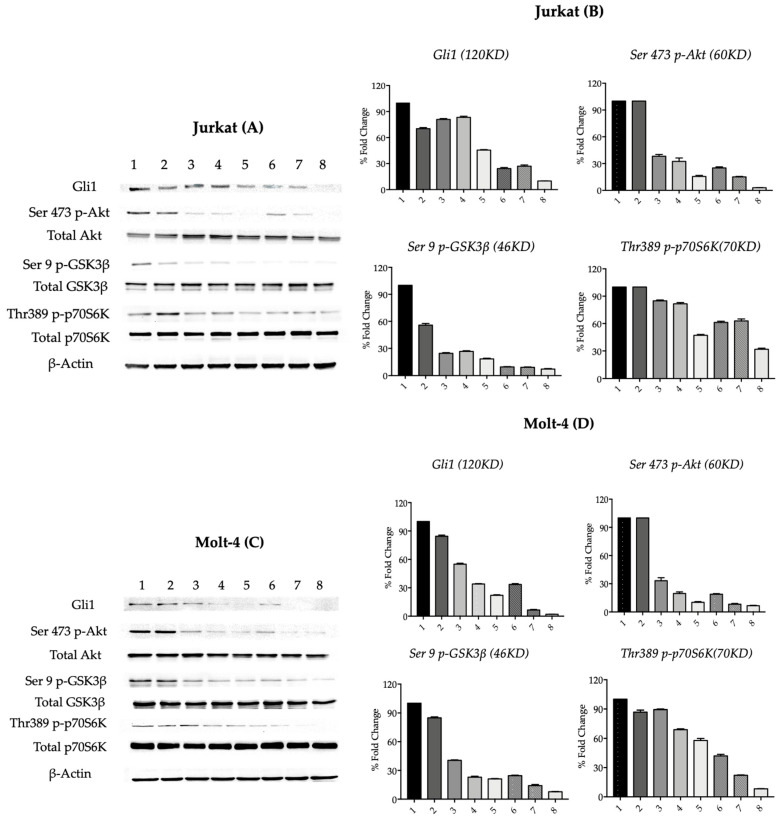
Western blot analysis of protein expression in Jurkat (**A**,**B**) and Molt-4 (**C**,**D**) cells. Both cell models were treated with GANT61 (lane 2), increasing concentrations of MK-2206 (lanes 3–5), and with the combination of GANT61 and MK-2206 at escalating concentrations (lanes 6–8). (**A**) shows protein expression in Jurkat cells (1: Ctrl, 72 h; 2: GANT61, 19 μM, 72 h; 3: MK-2206, 0.2 μM, 24 h; 4: MK-2206, 0.4 μM, 24 h; 5: MK-2206, 2 μM, 24 h; 6: GANT61 19 μM + MK-2206 0.2 μM; 7: GANT61 19 μM + MK-2206 0.4 μM; 8: GANT61 19 μM + MK-2206 2 μM). (**C**) shows the corresponding expression in Molt-4 cells (1: Ctrl, 72 h; 2: GANT61, 20 μM, 72 h; 3: MK-2206, 0.4 μM, 24 h; 4: MK-2206, 0.8 μM, 24 h; 5: MK-2206, 2.8 μM, 24 h; 6: GANT61 20 μM + MK-2206 0.4 μM; 7: GANT61 20 μM + MK-2206 0.8 μM; 8: GANT61 20 μM + MK-2206 2.8 μM). Total protein lysates were analyzed for the expression of the indicated proteins (Gli1; total and phosphorylated Akt, GSK3β, and p70S6K). β-actin was used as a loading control. (**B**,**D**) show the densitometric quantification of Western blot bands corresponding to Gli1, p-Akt, p-GSK3β, and p-p70S6K in Jurkat and Molt-4 cells, respectively. Densitometric analysis was performed only for proteins that exhibited labeling-related modifications. Band intensities were normalized to β-actin and expressed as fold change relative to the control. Data are presented as mean ± SD from three independent experiments.

**Figure 5 cells-14-01972-f005:**
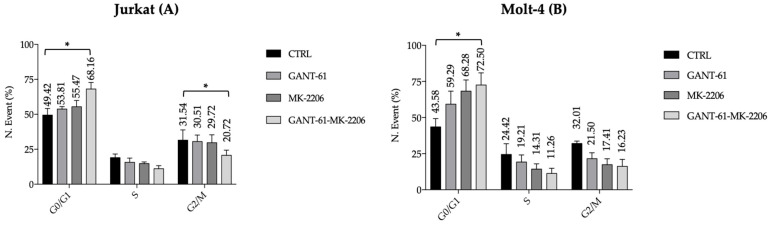
Cell cycle analysis in Jurkat (**A**) and Molt-4 (**B**) cells. Cell cycle distribution was assessed using the Muse Cell Cycle Kit. Treatments were performed at the respective IC_50_ concentrations of each inhibitor (Jurkat cell model: GANT-61, 19 μM for 72 h; MK-2206, 2 μM for 24 h; MOLT-4 cell model: GANT-61, 20 μM for 72 h; MK-2206, 2.8 μM for 24 h). Images are representative of three independent experiments. Data were analyzed using two-way ANOVA followed by multiple comparisons. In (**A**), an asterisk (*) indicates a statistically significant difference (*p* < 0.05) between the combined treatment and both the control and single treatments in the G_0_/G_1_ and G_2_/S phases. In (**B**), an asterisk (*) indicates a statistically significant difference (*p* < 0.05) between the combined treatment and the control in the G_0_/G_1_ phase.

**Figure 6 cells-14-01972-f006:**
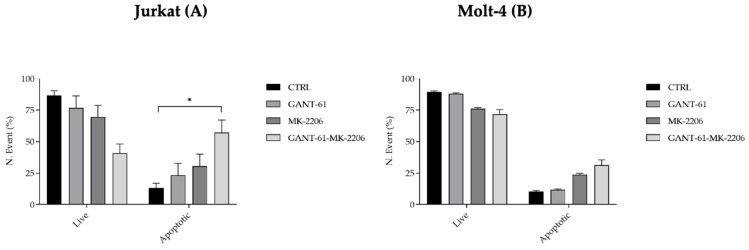
Apoptosis analysis in Jurkat (**A**) and Molt-4 (**B**) cells. Apoptosis was analyzed using the Muse Annexin V & Dead Cell Kit. All treatments were performed at the respective IC_50_ concentrations of each inhibitor (Jurkat cell model: GANT-61, 19 μM for 72 h; MK-2206, 2 μM for 24 h; MOLT-4 cell model: GANT-61, 20 μM for 72 h; MK-2206, 2.8 μM for 24 h). Images are representative of three independent experiments. Data were analyzed using two-way ANOVA followed by multiple comparisons. In (**A**), an asterisk (*) indicates a statistically significant difference (*p* < 0.05) between the combined treatment and both the control and single treatments.

**Figure 7 cells-14-01972-f007:**
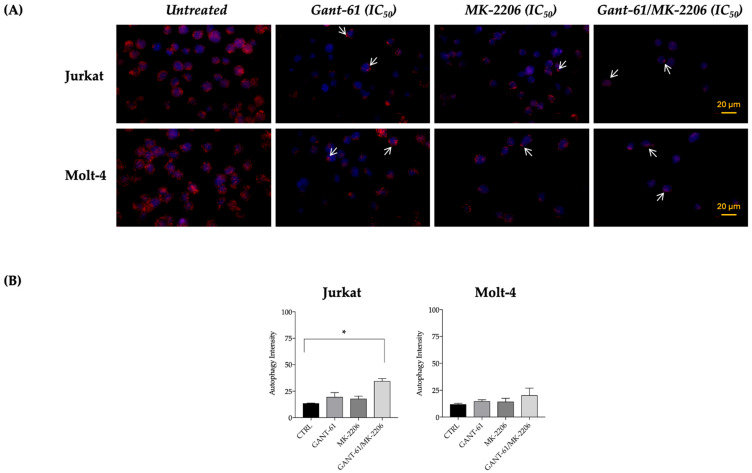
Autophagy analysis in Jurkat and Molt-4 T-cell leukemia cell lines. Representative immunofluorescence images showing LC3 localization to assess autophagy status in Jurkat and Molt-4 cells (**A**). Cells were either untreated or treated for 6 h and 24 h with GANT61 (IC_50_: 19 μM in Jurkat and 20 μM in Molt-4) or MK-2206 (IC_50_: 0.7 μM in Jurkat and 1.8 μM in Molt-4). Nuclei were stained with DAPI (blue). The scale bar is shown in the last image of the panel. The white arrows indicate the location of the spots. Quantification of autophagic activity in Jurkat and Molt-4 cells using the Muse Autophagy LC3-antibody–based Kit (**B**). LC3-II levels were measured following treatment with each inhibitor at its respective IC_50_ concentration (Jurkat: GANT61 19 μM, 48 h; MK-2206 2 μM, 48 h; Molt-4: GANT61 20 μM, 48 h; MK-2206 2.8 μM, 48 h). Asterisk (*) indicates a statistically significant difference (*p* < 0.05) between the combined treatment and the control Images and data are representative of three independent experiments.

**Table 1 cells-14-01972-t001:** IC_50_ values of MK-2206 and Gant-61 in T-ALL cell lines. IC_50_ values (μM) of the Akt inhibitor MK-2206 after 24 and 48 h of treatment and Hedgehog pathway inhibitor Gant-61 after 72 h of treatment were determined in Jurkat, Molt-4, DND-41, and ALL-SIL T-ALL cell lines. Data are presented as the mean ± SD from at least three independent experiments.

Cell Lines IC_50_ (μM)	RAD-001 (24–48 h)	MK-2206 (24 h)	MK-2206 (48 h)	Gant-61 (72 h)
Jurkat	Not Available	0.7 ± 0.3	0.6 ± 0.4	19 ± 5
Molt-4	Not Available	1.8 ± 0.6	0.5 ± 0.3	20 ± 2
DND-41	Not Available	15.6 ± 3	7.9 ± 6	24 ± 1
ALL-SIL	Not Available	12.7 ± 4	4.3 ± 3.5	16 ± 1

## Data Availability

The original contributions presented in this study are included in the article. Further inquiries can be directed to the corresponding author.

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
