# Peer review of "Dual Hedgehog/GLI1 and PI3K/Akt/mTOR Targeting Possesses Higher Efficacy to Inhibit T-Cell Acute Lymphoblastic Leukemia Growth"

_cells, 2025, doi:10.3390/cells14241972_

Round 1

Reviewer 1 Report

Comments and Suggestions for Authors

The authors have investigated the dual targets of the Gli and ALK pathways in T-ALL and have presented a promising therapy. Still, there are some issues that need to be addressed.

Major concerns

  1. The data show a good correlation between these targets' inhibitions; the mechanism through which this is achieved is not demonstrated. Some knockdown experiments should be done.
  2. There are no quantitative analyses performed, like densitometry for Western blot, cell cycle phase percentage; this should be included.
  3. The conclusion claims therapeutic promise, but no normal cell controls or primary patient samples are tested. At a minimum, include a discussion about selectivity and potential off-target effects. Reference whether the Gant-61 and MK-2206 doses used are achievable in vivo.

Minor concerns

  1. The manuscript has some English grammar and syntax errors that need to be addressed.
  1. Some figure labels are missing in the figures, and are present only in the text ( like Figures 1 and 7).
  1. Verify the reference to be consistent with the Cells style.

Author Response

Reviewer 1- Major concerns

Comment 1: The data show a good correlation between these targets' inhibitions; the mechanism through which this is achieved is not demonstrated. Some knockdown experiments should be done.

Response 1: Thank you for your valuable feedback. In the present study, we did not focus on knockdown experiments because the inhibitors used, MK-2206 and GANT-61, are well-established in the literature for their high specificity and well-characterized mechanisms of action.

MK-2206 is a potent allosteric inhibitor of AKT1/2/3, preventing AKT phosphorylation at T308 and S473 and consequently blocks AKT activation downstream of PI3K, thereby blocking its role in cell survival and proliferation (see for example: Hirai, H et al 2010; Uko, N. E et al 2020; Xing, Y et al 2019; Li, Z. et al 2012).

GANT-61 directly targets Gli1 and Gli2, the final transcriptional effectors of the Hedgehog signaling pathway, inhibiting their DNA-binding activity and transcriptional activation of target genes (see for example: Tong, W., et al 2018; Mazumdar, T et al 2011).

Given the extensive previous validation of these compounds’ specificity and mechanisms, we considered pharmacological inhibition a reliable and mechanistically relevant approach to investigate the functional crosstalk between these signaling pathways in our model.

We have clarified this point in the revised manuscript

Introduction: page 3 line 70-78

Comment 2: There are no quantitative analyses performed, like densitometry for Western blot, cell cycle phase percentage; this should be included.

Response 2: We thank the reviewer for this valuable comment. Following the valuable suggestion of the reviewer, we have modified the cell cycle analysis figure including the percentage values of the different cell cycle phases (Fig5). In addition, we have performed the densitometric analysis of the Western blot data, that showed expression changes which is now presented in figures 4 (Jurkat A-B; Molt-4 C-D).

Comment 3: The conclusion claims therapeutic promise, but no normal cell controls or primary patient samples are tested. At a minimum, include a discussion about selectivity and potential off-target effects. Reference whether the Gant-61 and MK-2206 doses used are achievable in vivo.

Response 3: We thank the reviewer for this comment. We acknowledge that the absence of normal cell controls and primary patient samples may represents a limitation of the present study. Our primary aim was to investigate the effect of simultaneously drug treatment of AKT and Hedgehog/GLI pathways. To achieve this goal, we employed established leukemia cell lines as models for checked and reproducible experiments, pharmacokinetic studies have demonstrated that plasma concentrations in the low micromolar range comparable to the doses used in our experiments are achievable in vivo and have been safely administered in clinical trials (see for example: Larsen, J. T. et al 2017).

Although GANT-61 is primarily used in preclinical studies, several reports indicate that effective concentrations, are within the pharmacologically achievable window in vivo (see for example: Gonnissen, A., et al 2013).

We have clarified this point in the revised manuscript

Discussion: page 17 lines 384-391

                      pag 18 lines 412-415

Reviewer 1-Minor concerns

Comment 1: The manuscript has some English grammar and syntax errors that need to be addressed.

Response 1: We appreciate the reviewer’s suggestion. We revised the entire manuscript.

Comment 2: Some figure labels are missing in the figures and are present only in the text (like Figures 1 and 7).

Response 2: We thank the reviewer for this helpful comment. We have now added the missing labels and legends to the figures accordingly.

Comment 3: Verify the reference to be consistent with the Cells style.

Response 3: We thank the reviewer for the observation. We have revised all the references to ensure they are consistent with theCells journal style.

Reviewer 2 Report

Comments and Suggestions for Authors

The manuscript "Dual Hedgehog/GLI1 and PI3K/Akt/mTOR targeting has higher efficacy to inhibit T-Cell Acute Lymphoblastic Leukemia growth" written by De Chiara M, Sicurella M, Conti I, Melloni M and Neri LM, presents the results of the treatment of several T-ALL cell lines with inhibitors of Hedgehog and PI3K pathways and their combination.  Processes analyzed are cell viability and drug cytotoxicity, expression of Gli1 and phosphorylated Akt, as well as some other PI3K downstream molecules, cell cycle analysis and appearance of apoptosis and autophagy.

When presenting T ALL cell lines, it would be more informative if also the status of their PI3K and Hh pathways is described.

Autophagy was analyzed by immunostaining, and in the Materials and methods another method was described (Muse autophagy LC3-antibody based kit). Decrease in LC3 expression detected by immunostaining was obtained as a result. LC3 is degraded only after fusion with lysosome, and on immunofluorescence analysis usually LC3 puncta are detected. The authors mention fluorescence staining in spots located in the cytoplasm. These spots should be signed and better explained. It is possible to detect LC3 on Western blot, or use some other methods to have clearer results.

The Discussion could be improved by discussing the use of mentioned inhibitors in cancer and leukemia therapy, clinical trials, in vivo experiments, dependence on the presence of leukemia cell mutations, etc. Some previous experiments using similar combinations are not mentioned (Hou, Biochimie, 2014). Explanation of non-canonical Smo-independent mechanism of Hh pathway activation can be given.

Other comments

Figure 1 graphs need bigger labels.

In Figure 6 legend method used for experiment is missing. On both, Fig 5. and Fig. 6, the meaning of the asterisk can be mentioned. Also, IC50 concentrations can be written. On Fig. 3, 6, 7. results should not be discussed in figure legends.

Author Response

Reviewer 2

The manuscript "Dual Hedgehog/GLI1 and PI3K/Akt/mTOR targeting has higher efficacy to inhibit T-Cell Acute Lymphoblastic Leukemia growth" written by De Chiara M, Sicurella M, Conti I, Melloni M and Neri LM, presents the results of the treatment of several T-ALL cell lines with inhibitors of Hedgehog and PI3K pathways and their combination. Processes analyzed are cell viability and drug cytotoxicity, expression of Gli1 and phosphorylated Akt, as well as some other PI3K downstream molecules, cell cycle analysis and appearance of apoptosis and autophagy.

Comment 1: When presenting T ALL cell lines, it would be more informative if also the status of their PI3K and Hh pathways is described.

Response 1: We thank the reviewer for this helpful comment. We agree that including information on the activation status of thePI3K/AKT and Hedgehog (Hh)/GLI pathways in the T-ALL cell lines used is helpful to better understand the significance of our study. Specifically, the four T-ALL cell lines employed Jurkat, Molt-4, DND-41, and ALL-SIL are reported in the literature to have constitutive aberrant activation of the PI3K/AKT pathway. Regarding the Hedgehog/GLI pathway, previous studies have shown that GLI1 and GLI2 are relevantly expressed and transcriptionally active in several T-ALL models, including Jurkat and MOLT-4.

We have clarified this point in the revised manuscript

Introduction: page 2 lines 53-60

Comment 2: Autophagy was analyzed by immunostaining, and in the Materials and methods another method was described (Muse autophagy LC3-antibody based kit). Decrease in LC3 expression detected by immunostaining was obtained as a result. LC3 is degraded only after fusion with lysosome, and on immunofluorescence analysis usually LC3 puncta are detected. The authors mention fluorescence staining in spots located in the cytoplasm. These spots should be signed and better explained. It is possible to detect LC3 on Western blot or use some other methods to have clearer results.

Response 2: We thank the reviewer for this insightful comment. Autophagy was analyzed both by immunofluorescence and by flow cytometry, as described in the Materials and Methods section. In the immunofluorescence images, we have now highlighted the LC3 spot as suggested (Figure 7A) and clarified this aspect in the revised version of the manuscript. The results obtained from the Muse analysis are presented in Figure 7B.

We have clarified this point in the revised manuscript

Materials and Methods: page 4 lines 124-125

                                            pag 5 line 152

Comment 3: The Discussion could be improved by discussing the use of mentioned inhibitors in cancer and leukemia therapy, clinical trials, in vivo experiments, dependence on the presence of leukemia cell mutations, etc. Some previous experiments using similar combinations are not mentioned (Hou, Biochimie, 2014). Explanation of non-canonical Smo-independent mechanism of Hh pathway activation can be given.

Response 3:

We thank the reviewer for their constructive comments.

We have expanded the Discussion to provide more detailed information on the clinical and preclinical development of the inhibitors used in our study. We added references to MK-2206, emphasizing its pharmacological relevance in vivo, and further elaborated on the therapeutic potential of Gli1 inhibitors, such as GANT-61, within mouse models and emerging oncological applications.

We also mentioned the results reported by Hou et al. (Biochimie, 2014) to highlight how our work  is innovative since, our results give strength explore new cell lines, applies new drugs, analyse new cell process such as autophagy and cell cycle and new molecules of the signaling pathway with both western blotting and in situ immunofluorescence. The biological rationale for combined strategies and we expanded the section on non-canonical Hedgehog activation. Furthermore, the discussion now offers a more explicit assessment of the role of mutations in PI3K, Akt, TP53, or SUFU. We have also emphasized the importance of future stratified analyses based on patient-specific mutational profiles to better identify subgroups that could derive the greatest therapeutic benefit.

We have clarified this point in the revised manuscript

Discussion: page 16 lines 399-409

                      pag 17 lines 410-415

Reviewer 2-Other comments

Comment 4: Figure 1 graphs need bigger labels.

Response 4: We thank the reviewer for this helpful comment. We have increased the labels in Figure 1 to improve clarity and readability. Please note that increased labels are observable in the single image uploaded in the journal platform but become reduced when inserted in the merged file by the journal platform itself.

Comment 5: In Figure 6 legend method used for experiment is missing.

Response 5: We thank the reviewer for this comment. We have now included the description of the method used in the Figure 5 and 6 legend.

Comment 6: On both, Fig 5. and Fig. 6, the meaning of the asterisk can be mentioned. Also, IC50 concentrations can be written

Response 6: We thank the reviewer for this helpful comment. We have added an explanation of the asterisks in the figure legends 5 and 6 and included the ICâ‚…â‚€ values as suggested.

Comment 7: On Fig. 3, 6, 7. results should not be discussed in figure legends.

Response 7: We thank the reviewer. We included brief descriptions of the results in the figure legends to facilitate the reader’s understanding but following the reviewer’s suggestion we have removed these brief result comments.

Round 2

Reviewer 1 Report

Comments and Suggestions for Authors

-No comments

Author Response

Comments and Suggestions for Authors

-No comments

Reviewer 2 Report

Comments and Suggestions for Authors

The authors of the manuscript „"Dual Hedgehog/GLI1 and PI3K/Akt/mTOR targeting has higher efficacy to inhibit T-Cell Acute Lymphoblastic Leukemia growth" responded to the comments and improved the manuscript.

Minor comments:

Preformulation of the sentences needed: lines 180-182

225, 245-6

Author Response

Comment 1: Preformulation of the sentences needed: lines 180-182 225, 245-6

Response 1: We appreciate the reviewer’s suggestion. We have modified lines 180-182, 225, and 245-246 as requested.
